# Evaluation of Endocervical Curettage in Colposcopy in the Turkish Cervical Cancer Screening Program

**DOI:** 10.3390/jcm13154417

**Published:** 2024-07-28

**Authors:** Utku Akgor, Nejat Ozgul, Ali Can Gunes, Murat Turkyılmaz, Murat Gultekin

**Affiliations:** 1Division of Gynecologic Oncology, Department of Obstetrics and Gynecology, Faculty of Medicine, Hacettepe University, Ankara 06100, Turkey; 2Department of Obstetrics and Gynecology, Mamak State Hospital, Ankara 06320, Turkey; acgunes@hacettepe.edu.tr; 3Department of Cancer Control, Turkish Ministry of Health, Public Health Institute, Ankara 06200, Turkey

**Keywords:** endocervical curettage, colposcopy, cervical intraepithelial neoplasia, cervical cancer

## Abstract

**Background/Objectives:** To investigate the risk factors for CIN2+ lesions (cervical intraepithelial neoplasia 3 or worse) in endocervical curettage (ECC) and to evaluate the relationship between the addition of ECC to punch biopsy in terms of the yield of CIN2+ lesions. **Methods:** Between February 2018 and 2019, data on colposcopy results from 11,944 patients were gathered from the Cancer Department of the Turkish Ministry of Health across the country. A total of 6370 women whom were referred to colposcopy were included in this study. Risk factors were identified using both univariate and multivariate logistic analyses. **Results:** The median age was 42 years old (range, 30–65). ASC-H (atypical squamous cells-suggestive of high-grade squamous intraepithelial lesion)/HSIL (high-grade intraepithelial lesion) cytology (OR 7.648 95% CI (3.933–14.871)) and HPV (human papillomavirus)-16/18 infection (OR 2.541 95% CI (1.788–3.611)) were identified as risk factors for having CIN2+ lesions. CIN2+ diagnostic yield by ECC is only 1.2% all patients. CIN2+ diagnostic yield by punch biopsy and ECC are 9.7% and 6% of patients, respectively. A higher CIN2+ yield by ECC was observed with increasing age. Among cytology groups, ASC-H/HSIL has highest CIN2+ yield by ECC. Finally, in patients with incomplete visualization of the squamocolumnar junction (SCJ), ECC yields approximately twice as many CIN2+ lesions. **Conclusions:** ECC should be considered in cases of advanced patient age and in situations where the SCJ is not routinely visualized. In addition, evaluation of the endocervical canal is necessary in HPV-positive cases infected with HPV-16/18 types and in cases infected with HPV of any type but with cytological abnormalities.

## 1. Introduction

Cervical cancer is the fourth most frequently diagnosed cancer among women globally, with over 600,000 new cases and nearly 350,000 deaths annually [1]. It is well known that persistent infection with high-risk human papillomavirus (HPV) types that cause cancer is the main cause of the development of precancerous lesions known as cervical intraepithelial neoplasia (CIN), and cervical cancer. Widespread standardization and promotion of cervical cancer screening has enabled early detection and treatment of cervical cancer and precancerous lesions, leading to significant reductions in both incidence and mortality. Women with abnormal HPV DNA and cervical cytology testing results should be referred for further investigation [2].

Cervical colposcopy plays a crucial role in the evaluation of women with abnormal cervical screening results. It is frequently used to detect and assess cervical cancer and/or precursor lesions called cervical intraepithelial neoplasia (CIN) through biopsies taken from lesions. Colposcopy is very useful for detecting lesions on the surface of the cervix, but has limitations in assessing the cervical epithelium at risk within the endocervical canal. Although endocervical curettage (ECC) is an effective method of evaluating CIN and malignancy within the cervical canal, there are no established standards for ECC between national and international organizations. This lack of standardization leads to considerable variation in recommendations between guidelines [3,4]. Some organizations do not recommend ECC at all, some recommend its use under certain conditions, and others either do not address it or provide ambiguous guidance [5,6,7].

The most comprehensive and up-to-date recommendations for the use of ECC are based on the 2017 ECC guidelines from the American Society for Colposcopy and Cervical Pathology (ASCCP) [8], which had delayed ECC standards. In an effort to address this gap in the 2017 guidelines, the ASCCP working group has updated the standards for ECC in the context of colposcopy. The consensus provides recommendations for ECC based on a number of parameters, including cervical cytology results, HPV infection status, squamocolumnar junction (SCJ) assessment, and age.

The aim of our study was to investigate the risk factors for CIN2+ lesions (CIN3 or worse) in ECC and to evaluate the relationship between the addition of ECC to punch biopsy in terms of the yield of CIN2+ lesions, according to current guidelines. To the best of our knowledge, our study has the largest number of cases among the limited number of studies that have investigated this issue.

## 2. Materials and Methods

### 2.1. Study Design

In Turkey, women aged 30–65 years are invited for HPV-based screening every five years by primary health care providers, including general practitioners and KETEM screening centers. These screening services are provided free of charge to eligible individuals. HPV DNA sample collection kits (Qiagen HC2) are sent from one of two central laboratories located in Ankara or Istanbul. Two samples are collected from each woman, enabling cytology testing for those who test positive for HPV without requiring a separate visit. The first sample is collected using a brush and is transferred to a glass slide for conventional cytology. Asymptomatic screening individuals exhibiting HPV-16/18 positivity or abnormal smear results with any high-risk HPV (HR-HPV) positivity will be referred to colposcopy centers. Colposcopies performed in this study were also conducted at the discretion of the clinician, beyond the specified indications.

Colposcopic examinations were conducted by specialists in Obstetrics and Gynecology. Cervical biopsies were taken from any lesion suspected of CIN that was colposcopically directed. Endocervical curettage was typically applied based on the clinician’s decision in cases where the transformation zone of the cervical lesion or its proximal extension could not be adequately visualized. Endocervical curettage was performed using an endocervical curette, deviating from routine local practice, and processed as a histopathological specimen.

This is a retrospective cohort study designed by collecting data on colposcopy results of 11,944 patients nationwide, obtained from the Ministry of Health Cancer Department, between February 2018 and February 2019. The “Colposcopy Evaluation Form”, developed by the Department of Obstetrics and Gynecology at Hacettepe University, Turkish Society for Colposcopy and Cervical Pathology, and the Ministry of Health Cancer Department, was employed during data collection. This form includes the clinicopathological characteristics of all eligible women, including age, cervical cytology, HPV status, colposcopic findings such as transformation zone (TZ), procedures performed, and pathology results. Colposcopy-performing physicians completed these forms comprehensively and submitted them to the Ministry of Health Cancer Department every three months.

The study included a specific asymptomatic screening population of 11,994 HPV-positive patients. Due to the study design, several groups were excluded: 3756 patients who underwent colposcopy without biopsy, 572 patients who did not undergo colposcopy, 490 patients with a history of treatment for CIN or cervical cancer, and 431 patients with a cervical mass and/or abnormal vaginal bleeding. Of the 6883 patients who underwent colposcopic biopsy, the following groups were excluded 380 patients who underwent a ‘see and treat’ procedure such as hysterectomy, loop electrosurgical excision, or conization; 79 patients with AGC (atypical glandular cells) or AIS (adenocarcinoma in situ) cytology; and 54 patients with incomplete or inconsistent data (Figure 1). Approval for the study was granted by the Non-interventional Clinical Research Ethics Board of Hacettepe University under approval number GO 20/60. The necessity for written informed consent was obtained due to the retrospective and observational nature of the study.

### 2.2. Cytological Evaluation

Utilizing the Bethesda system, cytological evaluation yielded the subsequent classifications: NILM (negative for intraepithelial lesions or malignancy—indicating normal), ASC-US (atypical squamous cells—undetermined significance), ASC-H (atypical squamous cells—suggestive of high-grade squamous intraepithelial lesion), LSIL (low-grade intraepithelial lesion), HSIL (high-grade intraepithelial lesion), AGC, and others. Cytological results were categorized as normal, inadequate sampling, or abnormal. NILM and cases of infection were considered within the normal range.

### 2.3. HPV DNA Analysis

The collection of HPV DNA specimens was carried out using Hybrid Capture2 (Qiagen) kits. This test indicates the qualitative detection of 18 low-risk and high-risk types of HPV DNA (HPV types 6, 11, 16, 18, 31, 33, 35, 39, 42, 43, 44, 45, 51, 52, 56, 58, 59, and 68) in cervical specimens. Genotyping was performed using the CLART kit (Genomica). The analysis encompassed 35 low-risk, high-risk, and probably high-risk types of HPV DNA (HPV types 6, 11, 16, 18, 26, 31, 33, 35, 39, 40, 42, 43, 44, 45, 51, 52, 53, 54, 56, 58, 59, 61, 62, 66, 68, 70, 71, 72, 73, 81, 82, 83, 84, 85, and 89). For cases with negative results on the CLART kit, Rotor-Gene Q (QIAGEN’s real-time PCR cycler) was used to test for 15 types of HPV DNA (HPV types 16, 18, 31, 33, 35, 39, 45, 51, 52, 56, 58, 59, 66, 67, and 68). A comprehensive video presentation on sampling procedure was produced by the Ministry of Health at the start of the project. The link of the video https://www.youtube.com/watch?v=IBmAflRjI10 (accessed on 24 July 2024). None of the patients in the study tested negative for HPV. In cases where patients had multiple HPV types, each type was individually counted. HPV variants other than these 14 oncogenic types were collectively categorized as “HPV-other types (+)”. In cases where patients had multiple HPV types, each type was individually counted.

### 2.4. Statistical Analysis

All data were entered into and analyzed by the Statistical Package for the Social Sciences Windows Version 23.0 (SPSS, Chicago, IL, USA). We calculated chi-squared statistics with *p* values, and the cut-off for statistical significance was set at a *p* value of <0.05. Binary logistic regression was used to detect the risk factors. Odds ratios (ORs) and 95% confidence intervals (CIs) were calculated to assess the strength of any association.

## 3. Results

A total of 6370 women were included in our study. The median age was 42 years old (range, 30–65) with 41.8% (n = 2663) aged 30–39 years, 35% (n = 2227) aged 40–49 years, 18.9% (n = 1206) aged 50–59 years, and 4.3% (n = 274) aged 60–65 years. The most common cytology result was NILM (83.2%), followed by ASCUS (10.8), LSIL (4.5), ASC-H (1.3), and HSIL (0.2). Among all patients, those with HPV-16/18 (+), 12 non-16/18 HR-HPV (+), and HPV-other types (+), were distributed as 3047 patients (47.8%), 3017 (47.4%), and 306 (4.8%), respectively (Table 1).

In 68.5% (n = 4362) of patients, the SCJ was completely visualized. During the punch biopsy procedure, an additional ECC was performed in 64% (n = 4074) of all patients. In the ECC procedure, CIN 1 was observed in 8.8% (n = 359) of patients, CIN 2 in 1.8% (n = 76), and CIN 3 in 4.9% (n = 199). Twenty-one cases of cervical cancer were detected by the ECC procedure (0.5%). Of the pathology results from punch biopsies, 1499 (23.6%), 377 (5.9%), 559 (8.8%), and 59 (0.9%) patients were diagnosed with CIN1, CIN2, CIN3, and cervical cancer, respectively (Table 1).

Risk factors for detection of CIN2+ by ECC in univariate and multivariate Cox regression analysis are shown in Table 2. In univariate analysis, ASC-H or HSIL cytology was associated with a higher risk of having CIN2+ lesions than NILM (OR 7.702 95% CI (4.036–14.696)). Women with HPV-16/18 infection had a higher risk of CIN2+ than women without HPV-16/18 infection (OR 2.861 95% CI (2.184–3.748)). Patients whose SJC was partially or not visualized had a higher risk of having CIN 2+ lesions (OR 0.700 95% CI (0.534–0.968)). In the multivariable analysis, ASC-H/HSIL cytology (OR 7.648 95% CI (3.933–14.871)) and HPV-16/18 infection (OR 2.541 95% CI (1.788–3.611)) were identified as risk factors for having CIN2+ lesions (Table 2).

Table 3 shows ECC and punch biopsies in the detection of CIN 2+ lesions. CIN2+ diagnostic yield by ECC is 1.2% (n = 49) of 4074 patients. CIN2+ diagnostic yield by punch biopsy and ECC are 9.7% (n = 391) and 6% (n = 247) of all patients, respectively.

Of the 559 patients with CIN 3, 89 (15.9%) were recommended for follow-up. A total of 402 patients with CIN 3 (71.9%) underwent loop electrosurgical excision or conization. Of the 68 patients with CIN 3 (12.2% of the total), 45 patients (8.1%) underwent total abdominal hysterectomy, 18 patients (3.2%) underwent total laparoscopic hysterectomy, and 5 patients (0.9%) underwent vaginal hysterectomy. Follow-up data were not available for 57 patients with cervical cancer.

Figure 2 shows the ratios of CIN2+ lesions yielded by punch biopsy, punch biopsy plus ECC, and ECC alone. These ratios are stratified by age groups, HPV status, cytology results, and SCJ visualization. When evaluating the CIN2+ yield of the three methods within each stratified group, a higher CIN2+ yield by ECC was observed with increasing age. In the patient population stratified by HPV status, more CIN2+ lesions were detected by ECC in the HPV-16/18 group. In patients stratified by cytology result, the ASC-H/HSIL group shows the highest CIN2+ yield with ECC. Finally, in patients with incomplete visualization of the SCJ, ECC yields approximately twice as many CIN2+ lesions.

## 4. Discussion

In this study, ASC-H/HSIL cytology and HPV-16/18 infection were identified as significant risk factors for CIN2+ lesions detected by ECC, while different age groups and SCJ visualization were not found to be risk factors. A higher yield of CIN2+ with ECC alone was observed with increasing age, and ECC detected approximately twice as many CIN2+ lesions in patients with incomplete visualization of the SCJ. Looking at the yield of ECC in the two parameters where an increased risk was observed, a higher yield of CIN2+ with ECC alone was seen in the HPV-16/18 group and the highest yield of CIN2+ with ECC was seen in the ASC-H/HSIL group. Overall, in patients who underwent both punch biopsy and ECC, the additional diagnostic yield of CIN2+ with ECC alone was relatively low at only 1.2%.

Throughout life, the position of the SCJ shifts inwards and becomes less visible, particularly during perimenopause and menopause due to changes in hormone exposure, resulting in a significant decrease in SCJ visibility with age [9]. Therefore, the inability to fully visualize the entire transformation zone increases the risk of missing precancerous lesions and invasive cervical cancer. According to the ASCCP guidelines, ECC is recommended when the SCJ is not fully visible on colposcopy and is preferred in patients aged 40 years and older undergoing colposcopy, primarily to exclude the presence of an invisible lesion within the endocervical canal [10]. Benefit and cost-effectiveness studies also recommend ECC particularly for women over 50 [11,12]. In this study, more CIN2+ lesions were found on ECC when the SCJ was not or only partially visible than it was fully visualized (12.06% vs. 5.12%, respectively). In addition, similar to some previous studies, a marginal contribution of ECC was observed in older patients. This contribution ranged from 10% to 11% in patients aged 40–60 years, and 18.75% in patients over 60 years of age. Given that ECC resulted in a CIN2+ detection rate in patients over 60 years, the lack of ECC may lead to underdiagnosis. The present study is in line with current guidelines and many studies [11,12,13], highlighting the increasing role of ECC in older patients and in cases where the SCJ is not visible.

In our study, cervical cytology results were classified into three different risk groups according to the Bethesda system. Of these, only ASC-H/HSIL cytology is a significant predictor of CIN2+ lesions in ECC, whereas ASCUS/LSIL cytology is not a risk factor for CIN2+ lesions in ECC, regardless of HPV infection type. The yield of CIN2+ lesions by ECC alone was similar for both ASCUS/LSIL and ASC-H/HSIL cytology and are higher than for NILM cytology. Our results suggest that recommending ECC in cases of ASC-H/HSIL cytology may be beneficial and that ECC is an acceptable option for patients with ASCUS/LSIL cytology infected with any type of HPV. In contrast to our findings, Poomtavorn et al. concluded that ECC is unnecessary in women with ASC-US or LSIL without known HPV infection status due to the low risk of high-grade endocervical dysplasia [14]. Another recent study recommended ECC only for HSIL and ASC-H cytology compared to women with normal cytology [13]. In this study, the higher CIN2+ ECC yield in ASCUS/LSIL, which is inconsistent with the literature, may be due not only to the fact that all participants were infected with HPV, but also to the fact that the participants were over 30 years of age, an age group beyond the 20s when clearance is highest [15], and had been exposed to persistent HPV infection for a longer period of time.

It is widely acknowledged that persistent HR-HPV infection of the uterine cervix is a leading cause of invasive cervical cancer and its precursors. HPV-16 and HPV-18 infections carry the highest risk for developing CIN2+ lesions and are responsible for 70% of cervical cancers, with other HR-HPV types following in prevalence [16,17]. This study identified HPV-16/18 as a significant risk factor for CIN2+ lesions in ECC compared to other HPV types, with the diagnostic yield of CIN2+ by ECC alone being higher in women with HPV-16/18 infection. Our findings suggest that ECC may be particularly beneficial for patients infected with HPV-16/18. The ASCCP guidelines for ECC at colposcopy recommend ECC for all patients undergoing colposcopy with a known positive test for HPV types 16 or 18, based on moderate evidence or limited clinical benefit [10]. HPV-18 is known to be closely associated with endocervical adenocarcinoma, and detection of these cancers at a pre-invasive stage is challenging. Therefore, our findings are consistent with the Gynaecologic Oncology Society 2020 guidelines [18], which recommend ECC for patients with HPV-18 infection.

Overall, 6% of ECC procedures result in CIN 2+ lesions, and the additional diagnostic yield of ECC after punch biopsy is only 1.2%. Considering that lesions below CIN2+ are managed with surveillance according to current guidelines [19], ECC changes disease management in only 20% of patients diagnosed with CIN2+ by ECC alone. The additional yield of ECC in the detection of CIN2+ or high-grade CIN varies significantly depending on the indication for biopsy. Studies in women with high-grade cytology have shown that the yield of high-grade CIN with ECC can be as high as 11.9% and 14.7% [4,20]. Another study found the diagnostic yield of ECC to be 1.01%, which is comparable to our findings and suggests that ECC is most beneficial in cases with high-risk cytology and advanced age. Unlike our study, this study did not document the HPV status of the participants [21]. On the other hand, the literature shows that in cases of low-grade squamous intraepithelial lesion cytology, the additional detection rate with ECC is much lower, and is reported to be around 0.6% [22]. Some studies suggest that the likelihood of detecting high-grade cervical lesions increases with the number of cervical punch biopsies performed. Therefore, it is expected that lower CIN2+ yields might be observed with ECC. Although our study did not include an analysis of the number of cervical biopsies performed, it should be noted that part of the low CIN2+ yield observed with ECC could be attributed to this factor [23].

The strength of this study is its large sample size, which addresses the role of ECC in colposcopy, a topic for which there is limited research. Given the retrospective nature of this study, the possibility of bias in the data cannot be overlooked. The absence of patients under 30 years of age in this screening population, the performance of unnecessary colposcopies in some patients, and the selection of samples only from HPV-infected patients are limitations of this study. Lastly, for more than a decade, HPV vaccination has been recommended for adolescents aged 9–14 years as the primary method of preventing HPV infection and HPV-associated disease [24]. HPV vaccines are also thought to be effective in preventing the progression and recurrence of CIN lesions [25]. However, there is no national vaccination program for the participants in this study, and the lack of data on their HPV vaccination status is another limitation.

## 5. Conclusions

Our study highlights the importance of ECC in detecting CIN2+ lesions, particularly in older patients and in situations where the SCJ is not routinely visualized. The data show that HPV-16/18 infection is a significant risk factor for CIN2+ lesions, supporting the need for ECC in HPV-positive cases infected with HPV-16/18 types and in cases infected with any HPV type but with cytological abnormalities. Our findings are consistent with current guidelines and emphasize the need for targeted use of ECC to improve early detection and management of CIN and cervical cancer. This study also highlights the limitations of retrospective analyses, and future prospective studies are needed to further validate our findings and refine guidelines for the use of ECC in cervical cancer screening.

## Figures and Tables

**Figure 1 jcm-13-04417-f001:**
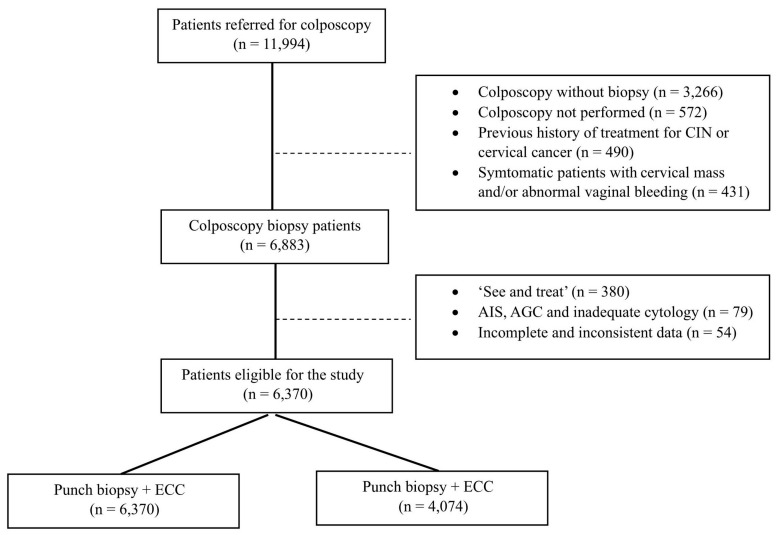
Flowchart of study population.

**Figure 2 jcm-13-04417-f002:**
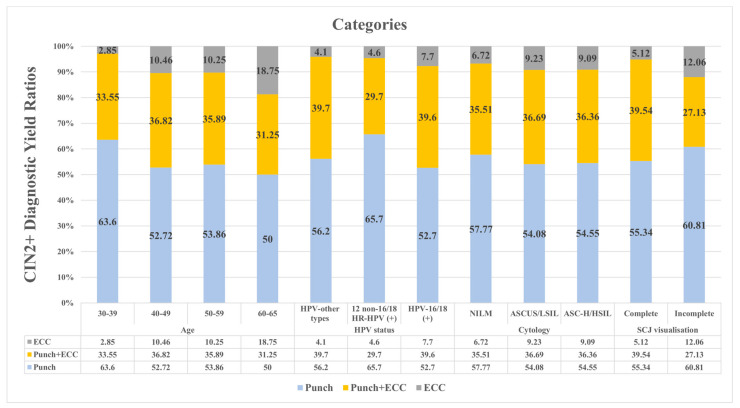
CIN2+ diagnostic yield rates in various categories.

**Table 1 jcm-13-04417-t001:** Demographic and clinical characteristics of study population.

Characteristics		N	%
Age			
	30–39	2663	41.8
	40–49	2227	35
	50–59	1206	18.9
	≥60	274	4.3
Cytology			
	NILM	5301	83.2
	ASC-US	688	10.8
	LSIL	288	4.5
	ASC-H	83	1.3
	HSIL	10	0.2
HPV status			
	HPV-16/18 (+) ^a^	3047	47.8
	12 non-16/18 HR-HPV (+) ^b^	3017	52.6
	HPV-other types (+) ^c^	306	4.8
SCJ Visualized			
	Complete	4362	68.5
	Partial or Not Visualized	2008	31.5
Procedure			
	Punch biopsy	2296	36
	Punch biopsy + ECC	4074	64
Pathology ECC			
	Normal	3419	84
	CIN 1	359	8.8
	CIN 2	76	1.8
	CIN 3	199	4.9
	Cervical cancer	21	0.5
Pathology punch			
	Normal	3878	60.8
	CIN 1	1499	23.6
	CIN 2	377	5.9
	CIN 3	559	8.8
	Cervical cancer	57	0.9

ASC-H, atypical squamous cell—cannot exclude HSIL; ASC-US, atypical squamous cell—undetermined significance; CIN, cervical intraepithelial neoplasia; ECC, endocervical curettage; HPV, human papillomavirus; HSIL, high-grade intraepithelial lesion; LSIL, low-grade intraepithelial lesion; NILM, negative for intraepithelial lesions or malignancy—normal or infection; SCJ, squamocolumnar junction. ^a^ Group labeled as HPV-16/18 (+) includes women with positive results only for HPV-16 and/or HPV-18. ^b^ Group labeled as 12 non-16/18 HR-HPV (+) includes women with positive results for the following HR-HPV genotypes: 31, 33, 35, 39, 45, 51, 52, 56, 58, 59, 66, and 68. ^c^ Group labeled as HPV-other types (+) includes women with positive results for other specific types: 6, 11, 26, 40, 42, 43, 44, 53, 54, 61, 62, 70, 71, 72, 73, 81, 82, 83, 84, 85, and 89.

**Table 2 jcm-13-04417-t002:** Risk factors of detecting CIN2+ in ECC.

Characteristics		Univariate		Multivariate	
		OR (95% CI)	*p* Value	OR (95% CI)	*p* Value
Age					
	30–39	1			
	40–49	1.082 (0.826–1.418)	0.567		
	50–59	0.995 (0.709–1.394)	0.975		
	≥60	0.718 (2.184–3.748)	0.382		
Cytology					
	NILM	1		1	
	ASC-US/LSIL	1.454 (0.993–2.127)	0.054		
	ASC-H/HSIL	7.702 (4.036–14.696)	<0.001	7.648 (3.933 14.871)	<0.001
HPV status					
	HPV-other types (+)	1		1	
	12 non-16/18 HR-HPV (+)	1.872 (0.770–5.690	0.219		
	HPV-16/18 (+)	2.861 (2.184–3.748)	<0.001	2.541 (1.788–3.611)	<0.001
SCJ Visualized					
	Complete	1		1	0.069
	Partial or Not Visualized	0.700 (0.534–0.968)	0.026	0.801 (0.745–1.105)	

ASC-H, atypical squamous cell—cannot exclude HSIL; ASC-US, atypical squamous cell—undetermined significance; HPV, human papillomavirus; HSIL, high-grade intraepithelial lesion; LSIL, low-grade intraepithelial lesion; NILM, negative for intraepithelial lesions or malignancy—normal or infection; SCJ, squamocolumnar junction.

**Table 3 jcm-13-04417-t003:** ECC and punch biopsies in detecting CIN 2+ lesions.

	Pathology ECC		
	≤CIN 2 (%)	CIN 2+ (%)	Total (%)
Pathology punch			
≤CIN 2 (%)	3387 (83.1)	49 (1.2)	3436 (84.3)
CIN 2+ (%)	391 (9.7)	247 (6)	638 (15.7)
Total (%)	3778 (93.8)	296 (7.2)	4074 (100)

CIN, cervical intraepithelial neoplasia. CIN2+ diagnostic yield by ECC is 1.2% of all patients. CIN2+ diagnostic yield by punch biopsy is 9.7% of all patients. CIN2+ diagnostic yield by punch biopsy and ECC is 6% of all patients.

## Data Availability

The raw data of this article will be sent to anyone by the first author, without undue reservation.

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
