# Peer review of "Evaluation of Endocervical Curettage in Colposcopy in the Turkish Cervical Cancer Screening Program"

_jcm, 2024, doi:10.3390/jcm13154417_

Round 1

Reviewer 1 Report

Comments and Suggestions for Authors

Overall, the study explained nicely on evaluation of endocervical curettage in colonoscopy on cervical cancer screening program in Turkish population.

1. Please do elaborate what is  cervical intraepithelial neoplasia? Some of the readers might not know the background information about CIN?

2. Include the information why are you using HPV infection status in the study? 

3. Introduction need to be improved some information  is missing. connection between the points is also missing? 

4. what will be the effect of CIN in HPV vaccinated people? Now a day physicians suggesting to ge vaccinate the kids/women from 11- 12 years of age irrespective of their family history.

5. Authors included more information related to the study in the discussion part. if you have some background information in the introduction it will be very interesting to the reader and also increase readers understanding. 

Author Response

Editor and Reviewer Comments:

Reviewer #1 (Comments to the Author): Reviewer #1: Reviewer
Reviewer Comment/ Question 1:

Overall, the study explained nicely on evaluation of endocervical curettage in colposcopy on cervical cancer screening program in Turkish population. Please do elaborate what is cervical intraepithelial neoplasia? Some of the readers might not know the background information about CIN?

Authors Response 1: Thank you very much for your valuable feedback. We have added a statement in lines 30-32 that cervical intraepithelial neoplasia is a precursor lesion to cervical cancer, as follows:

Line 34-35: the main cause of the development of precancerous lesions known as cervical intraepithelial neoplasia (CIN), and cervical cancer.

Reviewer Comment/ Question 2: Include the information why are you using HPV infection status in the study? 

Authors Response 2: This feedback is really very important. As you initially suggested, we have made the following changes to provide general information, noting that the persistence of high-risk HPV infection can lead to cervical cancer as follows;

Line 32-33: It is well known that persistent infection with high risk human papillomavirus (HPV) types that cause cancer

Reviewer Comment/ Question 3: Introduction need to be improved some information is missing. connection between the points is also missing? 

Authors Response 3: We completely agree with your comments. Therefore, a general informational paragraph has been added to the introduction section as follows:

Line 31-39: Cervical cancer is the fourth most frequently diagnosed cancer among women globally, with over 600,000 new cases and nearly 350,000 deaths annually1. It is well known that persistent infection with high risk human papillomavirus (HPV) types that cause cancer is the main cause of the development of precancerous lesions known as cervical intraepithelial neoplasia (CIN), and cervical cancer. Widespread standardisation and promotion of cervical cancer screening has enabled early detection and treatment of cervical cancer and precancerous lesions, leading to significant reductions in both incidence and mortality. Women with abnormal HPV DNA and cervical cytology testings should be referred for further investigation2.

Reviewer Comment/ Question 4: what will be the effect of CIN in HPV vaccinated people? Now a day physicians suggesting to ge vaccinate the kids/women from 11- 12 years of age irrespective of their family history.

Authors Response 4: Indeed, it is important to discuss the vaccine when talking about HPV and the diseases it causes. Therefore, in the discussion section, we have added information about the recommended age groups for vaccination, the effect of the vaccine on CIN lesions, and the lack of information about vaccination for our participants, as follows:

Line 313-318: Lastly, for more than a decade, HPV vaccination has been recommended for adolescents aged 9-14 years as the primary method of preventing HPV infection and HPV-associated disease 24. HPV vaccines are also thought to be effective in preventing the progression and recurrence of CIN lesions 25. However, there is no national vaccination program for the participants in this study, and the lack of data on their HPV vaccination status is another limitation.

Reviewer Comment/ Question 5: Authors included more information related to the study in the discussion part. if you have some background information in the introduction it will be very interesting to the reader and also increase readers understanding. 

Authors Response 5: You are right on this matter as well. Based on your feedback, we included a statement regarding HPV vaccines. Additionally, we found the conclusion section of the discussion to be somewhat weak, so we made some additions, as follows:

Line 313-318:

Lastly, for more than a decade, HPV vaccination has been recommended for adolescents aged 9-14 years as the primary method of preventing HPV infection and HPV-associated disease 24. HPV vaccines are also thought to be effective in preventing the progression and recurrence of CIN lesions 25. However, there is no national vaccination program for the participants in this study, and the lack of data on their HPV vaccination status is another limitation.

Line 319-328:

Our study highlights the importance of ECC in detecting CIN2+ lesions, particularly in older patients and in situations where the SCJ is not routinely visualized. The data show that HPV 16/18 infection is a significant risk factor for CIN2+ lesions, supporting the need for ECC in HPV-positive cases infected with HPV-16/18 types and in cases infected with any HPV type but with cytological abnormalities.  Our findings are consistent with current guidelines and emphasise the need for targeted use of ECC to improve early detection and management of CIN and cervical cancer. The study also highlights the limitations of retrospective analyses, and future prospective studies are needed to further validate our findings and refine guidelines for the use of ECC in cervical cancer screening.

Reviewer 2 Report

Comments and Suggestions for Authors

Dear Authors,

Your manuscript addresses an important topic in cervical cancer screening and provides valuable insights into the role of endocervical curettage (ECC) in detecting CIN2+ lesions. The large sample size and the comprehensive data collection from various regions enhance the reliability of the findings. The manuscript is well-structured, and the discussion provides a thoughtful analysis of the results in the context of existing literature.

By addressing the points raised below, the manuscript can be further improved to ensure clarity and consistency.

Comments:

1. Line 102-109: The description of Sub-section 2.3 is unclear. Please specify the names of the tests used for HPV detection and clearly state their performance. For instance, clarify if the HC2 assay is a screening test that provides results as HPV positive or HPV negative, and whether it enables individual HPV 16 and HPV 18 genotyping. What about with other HPV types in HC2 assay?

Line 106: The statement "genotype 73.7" is confusing. Please verify the accuracy of the genotypes identified by the CLART kit (Genomica). Also, please describe the performance of the CLART assay clearly.

2. Labels for "Non-16/18 HR-HPV (+)," and "HR-HPV (-)" are unclear. If all patients included in the study were HPV positive, the term "HR-HPV (-)" is inappropriate. Please use another term to accurately reflect that these are HPV positive subjects. Also, please explain the label of the other two categories in order to accurately reflect which HPV types were covered.

Suggested label changes:

"HPV-16/18 (+)a" is correct.

"Non-16/18 HR-HPV (+)b" could be "12 non-16/18 HR-HPV (+)."

"HR-HPV (-)c" could be "HPV-other types (+)."

 Below Table 1, provide appropriate explanations, such as:

aGroup labeled as HPV-16/18 (+) includes women with positive results only for HPV-16 and/or HPV-18.

 bGroup labeled as 12 non-16/18 HR-HPV (+) includes women with positive results for the following HR-HPV genotypes: 31, 33, 35, 39, 45, 51, 52, 56, 58, 59, 68, 73. Please verify that this list is correct. Why was the type 66 left out?

 cGroup labeled as HPV-other types (+) includes women with positive results for other specific types. Please provide an appropriate explanation. I can't suggest labels for this because the specific types covered by the HC2 assay and the CLART kit are unclear.

 3. Ensure consistent terminology throughout the manuscript after you adjust these names.

 4. Please clarify what is included under "CIN2+ lesions." If this includes CIN 2, CIN 3, and cervical cancer, make sure to specify this.

 5. Based on the content of the table, a more precise title could be: "Demographic and Clinical Characteristics of Study Population."

The first row of Table 1, "Median age at admission" does not seem necessary. It can be removed for clarity and consistency with the rest of the table. This data can be included in the text of the results.

 6. Figure 2: Ensure all axis labels and legend labels are clearly readable.

The Y-axis should be labeled: "CIN2+ Diagnostic Yield Ratios (%)."

The X-axis should be labeled: "Categories."

Figure 2 is not clearly visible. Consider using more intense colors and black font for the labels to enhance readability.

 7.  A comprehensive conclusion is needed. A strong conclusion not only summarizes the key findings but also highlights the implications of the study and suggests directions for future research. Re-summarize your key points emphasizing the importance of adding ECC to punch biopsies in certain patient groups. Also emphasize the relevance of your findings to clinical practice. It would benefit from emphasizing that future prospective studies are needed to further validate your findings and refine guidelines for the use of ECC in cervical cancer screening.

8. There are several instances of typographical errors and inconsistent formatting throughout the manuscript that need to be corrected. This includes inconsistent use of units and missing spaces.

Author Response

Reviewer Comment/ Question 2:

Labels for "Non-16/18 HR-HPV (+)," and "HR-HPV (-)" are unclear. If all patients included in the study were HPV positive, the term "HR-HPV (-)" is inappropriate. Please use another term to accurately reflect that these are HPV positive subjects. Also, please explain the label of the other two categories in order to accurately reflect which HPV types were covered.

Suggested label changes:

"HPV-16/18 (+)a" is correct.

"Non-16/18 HR-HPV (+)b" could be "12 non-16/18 HR-HPV (+)."

"HR-HPV (-)c" could be "HPV-other types (+)."

 Below Table 1, provide appropriate explanations, such as:

aGroup labeled as HPV-16/18 (+) includes women with positive results only for HPV-16 and/or HPV-18.

 bGroup labeled as 12 non-16/18 HR-HPV (+) includes women with positive results for the following HR-HPV genotypes: 31, 33, 35, 39, 45, 51, 52, 56, 58, 59, 68, 73. Please verify that this list is correct. Why was the type 66 left out?

 cGroup labeled as HPV-other types (+) includes women with positive results for other specific types. Please provide an appropriate explanation. I can't suggest labels for this because the specific types covered by the HC2 assay and the CLART kit are unclear.

Authors Response 2: Your criticisms and corrections are indeed very pertinent and we believe that your comments on terminology significantly improve the clarity of the manuscript. We fully agree with your suggestions and have implemented the corrections you have suggested, resulting in the following changes:

Line 115-131:

 2.3. HPV DNA Analysis

The collection of HPV DNA specimens was carried out using Hybrid Capture2 (Qiagen) kits. This test indicates the qualitative detection of 18 low-risk and high-risk types of HPV DNA (HPV types 6, 11, 16, 18, 31, 33, 35, 39, 42, 43, 44, 45, 51, 52, 56, 58, 59, and 68) in cervical specimens. Genotyping is performed using the CLART kit (Genomica). The analysis encompassed 35 low-risk, high-risk and probably high-risk types of HPV DNA (HPV types 6, 11, 16, 18, 26, 31, 33, 35, 39, 40, 42, 43, 44, 45, 51, 52, 53, 54, 56, 58, 59, 61, 62, 66, 68, 70, 71, 72, 73, 81 ,82, 83, 84, 85 and 89). For cases with negative results on the CLART kit, Rotor-Gene Q (QIAGEN's real-time PCR cycler) was used to test for 15 types of HPV DNA (HPV types 16, 18, 31, 33, 35, 39, 45, 51, 52, 56, 58, 59, 66, 67, and 68). A comprehensive video presentation on sampling procedure is being produced by the Ministry of Health at the start of the project. The link of the video https://www.youtube.com/watchv=ptplCnVtr7Q. None of the patients in the study tested negative for HPV. In cases where patients had multiple HPV types, each type was individually counted. HPV variants other than these 14 oncogenic types were collectively categorized as "HPV-other types (+)." In cases where patients had multiple HPV types, each type was individually counted.

Line 192-201;

Characteristics

N

%

    Age

30-39

2663

41.8

40-49

2227

35

50-59

1206

18.9

≥ 60

274

4.3

    Cytology

NILM

5301

83.2

ASC-US

688

10.8

LSIL

288

4.5

ASC-H

83

1.3

HSIL

10

0.2

    HPV status

HPV-16/18 (+)a

3047

47.8

12 non-16/18 HR-HPV (+)b

   3017

52.6

HPV-other types (+)c

306

4.8

    SCJ Visualized

Complete

4362

68.5

Partial or Not Visualized 

2008

31.5

    Procedure

Punch biopsy

2296

36

Punch biopsy + ECC

4074

64

    Pathology ECC

Normal

3419

84

CIN 1

359

8.8

CIN 2

76

1.8

CIN 3

199

4.9

Cervical cancer

21

0.5

    Pathology punch

Normal

3878

60.8

CIN 1

1499

23.6

CIN 2

377

5.9

CIN 3

559

8.8

Cervical cancer

57

0.9

Table1. Demographic and Clinical Characteristics of Study Population.

ASC-H, atypical squamous cell—cannot exclude HSIL; ASC-US, atypical squamous cell—undetermined significance; CIN, Cervical Intraepithelial Neoplasia; ECC, Endocervical Curettage; HPV, Human papillomavirus; HSIL, high-grade intraepithelial lesion; LSIL, low-grade intraepithelial lesion; NILM, negative for intraepithelial lesions or malignancy—normal or infection; SCJ, squamocolumnar junction.

aGroup labeled as HPV-16/18 (+) includes women with positive results only for HPV-16 and/or HPV-18.

bGroup labeled as 12 non-16/18 HR-HPV (+) includes women with positive results for the following HR-HPV genotypes: 31, 33, 35, 39, 45, 51, 52, 56, 58, 59, 66 and 68.

 cGroup labeled as HPV-other types (+) includes women with positive results for other specific types: 6, 11, 26, 40, 42, 43, 44, 53, 54, 61, 62, 70, 71, 72, 73, 81 ,82, 83, 84, 85 and 89.

Reviewer Comment/ Question 3: Ensure consistent terminology throughout the manuscript after you adjust these names.

Authors Response 3: Thank you for your suggestion. We have made the necessary changes in the text, tables, and figures accordingly. We have marked the changes with tracking and uploaded them to the system.

Reviewer Comment/ Question 4: Please clarify what is included under "CIN2+ lesions." If this includes CIN 2, CIN 3, and cervical cancer, make sure to specify this.

Authors Response 4: We referred to CIN2+ lesions as including CIN3 and more advanced lesions. We have clarified this in the introduction section as follows:

Line 59-60: The aim of our study was to investigate the risk factors for CIN2+ lesions (CIN3 or worse) in ECC and to evaluate the relationship between the addition of ECC to punch biopsy in terms of the yield of CIN2+ lesions, according to current guidelines.

Reviewer Comment/ Question 5: Based on the content of the table, a more precise title could be: "Demographic and Clinical Characteristics of Study Population."

The first row of Table 1, "Median age at admission" does not seem necessary. It can be removed for clarity and consistency with the rest of the table. This data can be included in the text of the results.

Authors Response 5: We agree with your comments. Accordingly, we have made the following changes:

Line 192 -193 ;

Table1. Demographic and Clinical Characteristics of Study Population.

Characteristics

N

%

    Age

30-39

2663

41.8

40-49

2227

35

50-59

1206

18.9

≥ 60

274

4.3

    Cytology

NILM

5301

83.2

ASC-US

688

10.8

LSIL

288

4.5

ASC-H

83

1.3

HSIL

10

0.2

    HPV status

HPV-16/18 (+)a

3047

47.8

12 non-16/18 HR-HPV (+)b

   3017

52.6

HPV-other types (+)c

306

4.8

    SCJ Visualized

Complete

4362

68.5

Partial or Not Visualized 

2008

31.5

    Procedure

Punch biopsy

2296

36

Punch biopsy + ECC

4074

64

    Pathology ECC

Normal

3419

84

CIN 1

359

8.8

CIN 2

76

1.8

CIN 3

199

4.9

Cervical cancer

21

0.5

    Pathology punch

Normal

3878

60.8

CIN 1

1499

23.6

CIN 2

377

5.9

CIN 3

559

8.8

Cervical cancer

57

0.9

Reviewer Comment/ Question 6: Figure 2: Ensure all axis labels and legend labels are clearly readable.The Y-axis should be labeled: "CIN2+ Diagnostic Yield Ratios (%)."

The X-axis should be labeled: "Categories."

Figure 2 is not clearly visible. Consider using more intense colors and black font for the labels to enhance readability.

Authors Response 6: Thank you for your comments. We agree with your comments on Figure 2. We have increased the font size and bolded the text in Figure 2. We have also labelled the X and Y axes as you suggested. The title of the table has been revised to read as follows:

Line.183-185;

Figure 2. CIN2+ diagnostic yield rates in various categories

Reviewer Comment/ Question 7: A comprehensive conclusion is needed. A strong conclusion not only summarizes the key findings but also highlights the implications of the study and suggests directions for future research. Re-summarize your key points emphasizing the importance of adding ECC to punch biopsies in certain patient groups. Also emphasize the relevance of your findings to clinical practice. It would benefit from emphasizing that future prospective studies are needed to further validate your findings and refine guidelines for the use of ECC in cervical cancer screening.

Authors Response 7: Thank you very much for your valuable contribution and comments. We agree that the conclusion section was quite weak. In line with your suggestions, we have made the following changes:

Line 320-328: Our study highlights the importance of ECC in detecting CIN2+ lesions, particularly in older patients and in situations where the SCJ is not routinely visualised. The data show that HPV 16/18 infection is a significant risk factor for CIN2+ lesions, supporting the need for ECC in HPV-positive cases infected with HPV-16/18 types and in cases infected with any HPV type but with cytological abnormalities.  Our findings are consistent with current guidelines and emphasise the need for targeted use of ECC to improve early detection and management of CIN and cervical cancer. The study also highlights the limitations of retrospective analyses, and future prospective studies are needed to further validate our findings and refine guidelines for the use of ECC in cervical cancer screening

Reviewer Comment/ Question 8: There are several instances of typographical errors and inconsistent formatting throughout the manuscript that need to be corrected. This includes inconsistent use of units and missing spaces.

Authors Response 8: We agree with you on this matter. Accordingly, we have made a series of changes throughout the manuscript, including the tables and figures. All changes have been highlighted in yellow.

Reviewer 3 Report

Comments and Suggestions for Authors

Thank you for your work concerning women’s health.

The paper is well-written. However, some issues needed to be revised.

-In introduction, please do not forget the vaginal bleeding symptom in endometrial cancer among postmenopausal women. Thus, this point should be excluded in this population by ultrasound. Nguyen PN, Nguyen VT. Endometrial thickness and uterine artery Doppler parameters as soft markers for prediction of endometrial cancer in postmenopausal bleeding women: a cross-sectional study at tertiary referral hospitals from Vietnam. Obstet Gynecol Sci. 2022;65(5):430-440. doi:10.5468/ogs.22053

-Inclusion and exclusion criteria should be more clarified.

-Study design should be defined as retrospective/prospective cohort/ cross-sectional study.

-Abbreviation word should not be placed at the first sentence of the paragraph. For example: Line 124: SCJ

-Figure 1 is not good at quality. Moreover, after patients eligible for the study (n=6.370), what were the next steps in this study?

-Abbreviation should be placed as the footnote under Table 2-3.

-Some baseline characteristics such as BMI, occupation, and demographic area should be added in Table 1.

-Some other high-risk types of HPV relating to cervical cancer were absent in Table 1. Please mention if the kit tests were absent at authors center.

-In statistical method, the study mention OR, but the result in Table 2 concerned HR.

-What the next steps the study did for 559 women of CIN 3 (conization) and 57 women of cervical cancer (surgical intervention: laparoscopic/laparotomic hysterectomy)?.

-Did the study could calculate the sensitivity/specificity/accuracy value of the tests used in this study compared to hallmark criteria by cytology/histology? I recommend this new study: Lalande E, Clarke H, Undurraga M, et al. Knowledge of cytology results affects the performance of colposcopy: a crossover study. BMC Womens Health. 2024;24(1):189. Published 2024 Mar 21. doi:10.1186/s12905-024-03025-y

-The study must be more interesting if the authors used the Kaplan-Meier analysis for the recurrence and development of higher-grade lesions after diagnosis.

-Reference citation did not follow the journal’s requirement. Line 177: Delete one “coma”.

-Please give the full-word for the abbreviation at the beginning of the paragraph. For example: The first graph of discussion: ASCCP, ASC-H/HSIL, ECC, SCJ, CIN2+.

Comments on the Quality of English Language

Minor editing of language is required.

Author Response

Reviewer #3 (Comments to the Author): Reviewer #3: Reviewer
Reviewer Comment/ Question 1:

Thank you for your work concerning women’s health.

The paper is well-written. However, some issues needed to be revised.

-In introduction, please do not forget the vaginal bleeding symptom in endometrial cancer among postmenopausal women. Thus, this point should be excluded in this population by ultrasound. Nguyen PN, Nguyen VT. Endometrial thickness and uterine artery Doppler parameters as soft markers for prediction of endometrial cancer in postmenopausal bleeding women: a cross-sectional study at tertiary referral hospitals from Vietnam. Obstet Gynecol Sci. 2022;65(5):430-440. doi:10.5468/ogs.22053

Authors Response 1:

Thank you for your feedback on the well-written paper. However, I would like to clarify that our study does not provide information on vaginal bleeding and endometrial cancer in the introduction. We understand that this comment may have been made unintentionally, and unfortunately we were unable to address it due to this oversight.

Reviewer Comment/ Question 2:

-Inclusion and exclusion criteria should be more clarified.

Authors Response 2: We noticed that the inclusion and exclusion criteria were not clearly and thoroughly described in the manuscript and in Figure 1. We have therefore added a detailed explanation of these criteria to the manuscript. This addition has significantly improved the quality of the paper. Thank you for bringing this to our attention. The changes are as below;

Line 95-103: The study included a specific asymptomatic screening population of 11,994 HPV-positive patients. Due to the study design, several groups were excluded: 3,756 patients who underwent colposcopy without biopsy, 572 patients who did not undergo colposcopy, 490 patients with a history of treatment for CIN or cervical cancer, and 431 patients with a cervical mass and/or abnormal vaginal bleeding. Of the 6,883 patients who underwent colposcopic biopsy, the following groups were excluded 380 patients who underwent a 'see and treat' procedure such as hysterectomy, loop electrosurgical excision or conisation; 79 patients with AGC (atypical glandular cells) or AIS (adenocarcinoma in situ) cytology; and 54 patients with incomplete or inconsistent data (Figure 1).

Reviewer Comment/ Question 3: Study design should be defined as retrospective/prospective cohort/ cross-sectional study.

Authors Response 3: We appreciate your valuable contribution. In response, we have now added a detailed description of the study methodology as follows:

Line 85-87: This is a retrospective cohort study designed by collecting data on colposcopy results of 11,944 patients nationwide, obtained from the Ministry of Health Cancer Department, between February 2018 and February 2019.

Reviewer Comment/ Question 4: Abbreviation word should not be placed at the first sentence of the paragraph. For example: Line 124: SCJ

Authors Response 4:

Thank you for reminding us of the general rule to avoid beginning a sentence with an acronym or an abbreviation. We made the relevant changes in the manuscript.

Line 146: In 68.5% (n=4362) of patients, the SCJ was completely visualized.

Reviewer Comment/ Question 5: Figure 1 is not good at quality. Moreover, after patients eligible for the study (n=6.370), what were the next steps in this study?

Authors Response 5: You are right in this regard. Following your suggestion, we have increased the quality of the figure to 300 dpi. Additionally, we have expanded the content as follows:

Line 181-182;

Reviewer Comment/ Question 6:

-Abbreviation should be placed as the footnote under Table 2-3.

Authors Response 6: Thank you for your important contribution. We have added abbreviations to Tables 2 and 3. The changes are as below;

Line 212-216: Table 2 abbreviations; ASC-H, atypical squamous cell—cannot exclude HSIL; ASC-US, atypical squamous cell—undetermined significance; HPV, Human papillomavirus; HSIL, high-grade intraepithelial lesion; LSIL, low-grade intraepithelial lesion; NILM, negative for intraepithelial lesions or malignancy—normal or infection; SCJ, squamocolumnar junction.

Line 225: Table 3 abbreviations; CIN, Cervical Intraepithelial Neoplasia.

Reviewer Comment/ Question 7: Some baseline characteristics such as BMI, occupation, and demographic area should be added in Table 1.

Authors Response 7:

Thank you for your suggestion. Unfortunately, our current dataset does not include demographic data of the patients (such as BMI, occupation, and other demographic information). Therefore, we regret that we are unable to implement your suggestion.

Reviewer Comment/ Question 8:

-Some other high-risk types of HPV relating to cervical cancer were absent in Table 1. Please mention if the kit tests were absent at authors center.

Authors Response 8: Thank you for your contribution. The second reviewer also raised a similar concern. Accordingly, we have made the following changes in the Materials and Methods section and Table 1:

Line 115-131:

 2.3. HPV DNA Analysis

The collection of HPV DNA specimens was carried out using Hybrid Capture2 (Qiagen) kits. This test indicates the qualitative detection of 18 low-risk and high-risk types of HPV DNA (HPV types 6, 11, 16, 18, 31, 33, 35, 39, 42, 43, 44, 45, 51, 52, 56, 58, 59, and 68) in cervical specimens. Genotyping is performed using the CLART kit (Genomica). The analysis encompassed 35 low-risk, high-risk and probably high-risk types of HPV DNA (HPV types 6, 11, 16, 18, 26, 31, 33, 35, 39, 40, 42, 43, 44, 45, 51, 52, 53, 54, 56, 58, 59, 61, 62, 66, 68, 70, 71, 72, 73, 81 ,82, 83, 84, 85 and 89). For cases with negative results on the CLART kit, Rotor-Gene Q (QIAGEN's real-time PCR cycler) was used to test for 15 types of HPV DNA (HPV types 16, 18, 31, 33, 35, 39, 45, 51, 52, 56, 58, 59, 66, 67, and 68). A comprehensive video presentation on sampling procedure is being produced by the Ministry of Health at the start of the project. The link of the video https://www.youtube.com/watchv=ptplCnVtr7Q. None of the patients in the study tested negative for HPV. In cases where patients had multiple HPV types, each type was individually counted. HPV variants other than these 14 oncogenic types were collectively categorized as "HPV-other types (+)." In cases where patients had multiple HPV types, each type was individually counted

Line 192-201: Table1. Demographic and Clinical Characteristics of Study Population.

Characteristics

N

%

    Age

30-39

2663

41.8

40-49

2227

35

50-59

1206

18.9

≥ 60

274

4.3

    Cytology

NILM

5301

83.2

ASC-US

688

10.8

LSIL

288

4.5

ASC-H

83

1.3

HSIL

10

0.2

    HPV status

HPV-16/18 (+)a

3047

47.8

12 non-16/18 HR-HPV (+)b

   3017

52.6

HPV-other types (+)c

306

4.8

    SCJ Visualized

Complete

4362

68.5

Partial or Not Visualized 

2008

31.5

    Procedure

Punch biopsy

2296

36

Punch biopsy + ECC

4074

64

    Pathology ECC

Normal

3419

84

CIN 1

359

8.8

CIN 2

76

1.8

CIN 3

199

4.9

Cervical cancer

21

0.5

    Pathology punch

Normal

3878

60.8

CIN 1

1499

23.6

CIN 2

377

5.9

CIN 3

559

8.8

Cervical cancer

57

0.9

ASC-H, atypical squamous cell—cannot exclude HSIL; ASC-US, atypical squamous cell—undetermined significance; CIN, Cervical Intraepithelial Neoplasia; ECC, Endocervical Curettage; HPV, Human papillomavirus; HSIL, high-grade intraepithelial lesion; LSIL, low-grade intraepithelial lesion; NILM, negative for intraepithelial lesions or malignancy—normal or infection; SCJ, squamocolumnar junction.

aGroup labeled as HPV-16/18 (+) includes women with positive results only for HPV-16 and/or HPV-18.

bGroup labeled as 12 non-16/18 HR-HPV (+) includes women with positive results for the following HR-HPV genotypes: 31, 33, 35, 39, 45, 51, 52, 56, 58, 59, 66 and 68.

 cGroup labeled as HPV-other types (+) includes women with positive results for other specific types: 6, 11, 26, 40, 42, 43, 44, 53, 54, 61, 62, 70, 71, 72, 73, 81 ,82, 83, 84, 85 and 89.

Reviewer Comment/ Question 9: In statistical method, the study mention OR, but the result in Table 2 concerned HR.

Reviewer Comment/ Question 9: Thank you for the correction. We are making the necessary adjustments to the table entry.

Line 211, table 2:

Univariate

Multivariate

OR (95% CI)

p value

OR (95% CI)

p value

Reviewer Comment/ Question 10: What the next steps the study did for 559 women of CIN 3 (conization) and 57 women of cervical cancer (surgical intervention: laparoscopic/laparotomic hysterectomy)?.

Authors Response 10:

Line 166-171; Of the 559 patients with CIN 3, 89 (15.9%) were recommended for follow-up. A total of 402 patients with CIN 3 (71.9%) underwent loop electrosurgical excision or conisation. Of the 68 patients with CIN 3 (12.2% of the total), 45 patients (8.1%) underwent total abdominal hysterectomy, 18 patients (3.2%) underwent total laparoscopic hysterectomy and 5 patients (0.9%) underwent vaginal hysterectomy. Follow-up data were not available for 57 patients with cervical cancer.

Reviewer Comment/ Question 11:

-Did the study could calculate the sensitivity/specificity/accuracy value of the tests used in this study compared to hallmark criteria by cytology/histology? I recommend this new study: Lalande E, Clarke H, Undurraga M, et al. Knowledge of cytology results affects the performance of colposcopy: a crossover study. BMC Womens Health. 2024;24(1):189. Published 2024 Mar 21. doi:10.1186/s12905-024-03025-y

Authors Response 11: We read the study you mentioned with great interest. It is a well-designed and valuable study. This study was designed to evaluate the extent to which knowledge of cytology results affects colposcopy interpretation. It supports the notion that known cytology influences the colposcopist's diagnosis. The main difference between this study and ours is that in our study, a large cohort with known cytology and HPV results is included. In contrast, the article you referenced included some patients without known cytology results, allowing the measurement of sensitivity for ≥CIN2 detection in the presence of cytology. In our dataset, only cytology results based on the Bethesda classification are available, and patients have not been evaluated according to Hallmark cytology criteria. However, upon reviewing the literature, we found that studies on Hallmark criteria and colposcopy are quite limited. We appreciate your highlighting this important topic and thank you for pointing it out for future research.

Reviewer Comment/ Question 12:The study must be more interesting if the authors used the Kaplan-Meier analysis for the recurrence and development of higher-grade lesions after diagnosis.

Authors Response 12: That's an excellent idea. Morever upon reviewing the literature, we did not find similar studies that describe Kaplan-Meier analysis for the recurrence and development of higher-grade lesions after diagnosis. Unfortunately, our current dataset does not contain this information. Therefore, we are unable to utilize this suggestion, despite its potential to greatly enhance our manuscript.

Reviewer Comment/ Question 13: Reference citation did not follow the journal’s requirement. Line 177: Delete one “coma”.

Authors Response 13 : Thank you for the correction. We now deleted the ‘’coma’’.

Reviewer Comment/ Question 14: Please give the full-word for the abbreviation at the beginning of the paragraph. For example: The first graph of discussion: ASCCP, ASC-H/HSIL, ECC, SCJ, CIN2+.

Authors Response : Thank you very much for your important warning on this matter. After considering your warning, we reviewed the Journal of Clinical Medicine's Instructions for Authors, specifically the guideline stating that "Acronyms/Abbreviations/Initialisms should be defined the first time they appear in each of three sections: the abstract; the main text; the first figure or table. When defined for the first time, the acronym/abbreviation/initialism should be added in parentheses after the written-out form." Following this guideline, we have made the necessary corrections throughout the entire manuscript. Your contribution is highly valuable and greatly appreciated. The changes are as below;

Line 10-27, Abstract:

Abstract: Background/Objectives: To investigate the risk factors for CIN2+ lesions (cervical intraepithelial neoplasia 3 or worse) in endocervical curettage (ECC) and to evaluate the relationship between the addition of ECC to punch biopsy in terms of the yield of CIN2+ lesions. Methods: Between February 2018 and 2019, data on colposcopy results from 11,944 patients were gathered from the Cancer Department of the Turkish Ministry of Health across the country. A total of 6370 women whom referred to colposcopy were included to this study. Risk factors were identified using both univariate and multivariate logistic analyses. Results: The median age was 42 years old (range, 30-65). ASC-H (atypical squamous cells-suggestive of high-grade squamous intraepithelial lesion) / HSIL (high-grade intraepithelial lesion) cytology (OR 7.648 95% CI (3.933-14.871)) and HPV (human papillomavirus) -16/18 infection (OR 2.541 95% CI (1.788-3.611)) were identified as risk factors for having CIN2+ lesions. CIN2+ diagnostic yield by ECC is only 1.2% all patients. CIN2+ diagnostic yield by punch biopsy and ECC are 9.7 % and 6% of patients, respectively. A higher CIN2+ yield by ECC was observed with increasing age. Among cytology groups ASC-H/HSIL has highest CIN2+ yield by ECC. Finally, in patients with incomplete visualisation of the squamocolumnar junction (SCJ), ECC yields approximately twice as many CIN2+ lesions. Conclusions: ECC should be considered in cases of advanced patient age and in situations where the SCJ is not routinely visualised. In addition, evaluation of the endocervical canal is necessary in HPV-positive cases infected with HPV-16/18 types and in cases infected with HPV of any type but with cytological abnormalities.

Line 34 -35 , Introduction:

It is frequently used to detect and assess cervical cancer and/or precursor lesions called cervical intraepithelial neoplasia (CIN) through biopsies taken from lesions.

Line 46 -47 , Introduction:

Although endocervical curettage (ECC) is an effective method of evaluating CIN and malignancy within the cervical cana

Round 2

Reviewer 3 Report

Comments and Suggestions for Authors

Dear authors,

Thank you for your revision.

The paper is well-improved now.

Best regards,